# Scanning electrochemical microscopy screening of CO₂ electroreduction activities and product selectivities of catalyst arrays

Francis D. Mayer[1], Pooya Hosseini-Benhangi [2,3], Carlos M. Sánchez-Sánchez [4], Edouard Asselin [5] & Előd L. Gyenge[1✉]

The electroreduction of $CO_2$ is one of the most investigated reactions and involves testing a large number and variety of catalysts. The majority of experimental electrocatalysis studies use conventional one-sample-at-a-time methods without providing spatially resolved catalytic activity information. Herein, we present the application of scanning electrochemical microscopy (SECM) for simultaneous screening of different catalysts forming an array. We demonstrate the potential of this method for electrocatalytic assessment of an array consisting of three $Sn/SnO_x$ catalysts for $CO_2$ reduction to formate (CO2RF). Simultaneous SECM scans with fast scan (1 V s⁻¹) cyclic voltammetry detection of products ($HCOO^-$, CO and $H_2$) at the Pt ultramicroelectrode tip were performed. We were able to consistently distinguish the electrocatalytic activities of the three compositionally and morphologically different $Sn/SnO_x$ catalysts. Further development of this technique for larger catalyst arrays and matrices coupled with machine learning based algorithms could greatly accelerate the $CO_2$ electroreduction catalyst discovery.

---

[1] Department of Chemical and Biological Engineering, Clean Energy Research Centre, The University of British Columbia, 2360 East Mall, Vancouver, BC V6T 1Z4, Canada. [2] Department of Materials Engineering, The University of British Columbia, 6350 Stores Road, Vancouver, BC V6T 1Z4, Canada. [3] Agora Energy Technologies Ltd., 3800 Wesbrook Mall, Vancouver, BC V6S 2L9, Canada. [4] Sorbonne Université, CNRS, Laboratoire Interfaces et Systèmes Electrochimiques, LISE, 75005 Paris, France. [5] Department of Materials Engineering, Canada Research Chair in Aqueous Processing of Metals, The University of British Columbia, 6350 Stores Road, Vancouver, BC V6T 1Z4, Canada. ✉email: elod.gyenge@ubc.ca

Aqueous formate salt solutions have been proposed as an easy to handle energy vector in renewable energy storage and conversion processes. In one approach, excess renewable electrical energy is used to electroreduce carbon dioxide to formate in aqueous media. The stored liquid formate solution is then electro-oxidized in either a direct formate fuel cell[1] or in a $CO_2$ redox flow battery[2] when and where energy is needed, while also contributing to a carbon neutral energy cycle. The electrochemical production of formate from $CO_2$ has been hampered by catalyst related issues such as low catalytic activity, durability and selectivity. One of the most promising catalysts for this process is $Sn/SnO_2$, which has shown high selectivity and low overpotential for CO2RF[3,4]. The adsorption and surface interaction of the radical anion $CO_2^{\bullet-}$ is a key step in the reaction mechanism influencing the overall reaction rate[5–12]. The nature of the catalytic active sites can include an active, nascent, Sn surface formed in situ by oxide reduction and the oxide itself[12]. It has been proposed that maintaining the stability of the surface tin oxide structure is important for the long-term catalytic durability in CO2RF[5,6].

The experimental variables influencing the performance of CO2RF catalysts are: catalyst material[7], co-catalyst[8], support (e.g., C-black vs. graphene), catalyst size and morphology[9], oxidation state[10], surface defect[11,12], and electrolyte composition[13]. Since these variables can be used in combination with one another, optimizing the catalyst performance will be a slow, tedious process if each catalyst formulation is experimentally tested one at a time in half-cell experiments using techniques such as linear or cyclic voltammetry at static and rotating disk/ring-disk electrodes (RDE/RRDE)[14,15]. Generally, a major hindrance for new electrocatalyst discovery is the lack of high-throughput experimental screening methods. On the theoretical side of catalyst discovery important advancements are made with accelerated machine learning based methods coupled with density functional theory computations[16]. The accelerated theoretical findings can guide the experimental studies in efficiently narrowing the range of investigated catalyst formulations. However, at present there are only a few multi-electrocatalyst screening methods that could be used. Optical screening methods are fast, but are limited to optically active reaction systems. Multi-electrode array cells might also be used, but their application to complex catalyst systems is limited by the array manufacturing process[17]. Scanning probe microscopy methods, such as scanning electrochemical microscopy (SECM) and the more complex scanning flow cell (SFM)[18], have had widespread application in electrochemistry, but none have been used to screen multiple CO2RF electrocatalysts at the same time. SECM has been used for screening of catalysts for a few reactions, such as hydrogen evolution, oxygen evolution, oxygen reduction[19] and formic acid oxidation[20–22]. SECM studies of CO2RF catalyst characterization were focused on individual carbon nanomaterials (graphene, nanotube) or precious metal catalysts (Ag, Au, Pd)[23–25]. Mirroring the RRDE technique[15], combinations of SECM and ultramicroelectrode (UME) tip CVs were used for in-situ rapid product analysis for single CO2RF catalyst[23–25]. To our knowledge, no report exists of this technique being extended to the simultaneous analysis of multiple CO2RF catalysts.

The goal of our work is the investigation of SECM as a product selective screening method of catalyst arrays for CO2RF using catalyst compositions and morphologies of practical relevance as opposed to model (e.g., smooth single crystal) surfaces. The focus here is on $Sn/SnO_x$ catalysts and their activity for CO2RF. Arrays composed of three $Sn/SnO_x$ surfaces were fabricated by a novel method based on different electroreduction pre-treatments of each sample, thus, each catalyst had different surface composition and morphology. We demonstrate that simultaneous

SECM scanning of the array gave consistent results proving the suitability of this technique for fast evaluation of electrocatalytic activities for different catalysts composing the array.

## Results

**Synthesis and characterizations of $Sn/SnO_x$ catalyst arrays.** Flat, mirror polished, Sn substrates with native oxides ($SnO_x$, $x = 1$ and 2) on the surface were prepared and used as catalyst precursors (Fig. 1a). The morphology of the polished substrate is typical of a heavily worked Sn surface; with a few identifiable shallow, oriented polishing scratches[26,27]. In order to produce different surface morphologies and compositions that are expected to have an effect on the CO2RF activity, the polished precursor substrate was modified by pre-electroreduction in a $N_2$ saturated 0.1 M $KHCO_3$ solution (pH 8.75) for 30 min at 293 K. Note the term pre-electroreduction is used to indicate the surface pre-treatment preceding the actual CO2RF experiment. The pre-electroreduction potentials of −1.25 V vs. Ag/AgCl and −3 V vs. Ag/AgCl were chosen based on surface pre-treatment screening experiments (Supplementary Fig. 1), in order to produce two types of morphologies: micro-scale spherical particles and nanoparticles, respectively. As shown by Fig. 1a, pre-electroreduction at −1.25 V (or more generally between −1 and −2 V, Supplementary Fig. 1) generated a roughened surface with some larger spherical aggregates with diameters from 100 nm to 200 μm, whereas pre-electroreduction at −3 V (or −3.5 V, Supplementary Fig. 1) produced a surface covered by spherical nanoparticles with diameters ranging from 30 to 70 nm. Nanoparticle formation by electroreduction has been previously reported for indium-tin oxide (ITO)[28]. We also observed similar nanoparticle surface coverage when pre-electroreduction at −3 V was applied to either chemically or electrochemically formed tin oxide (Supplementary Fig. 2). The mechanism for ITO nanoparticle formation during electroreduction was proposed to be due to dissolution-precipitation of the dissolved metallic ions[29]. We propose that a similar mechanism is at play here as well for the pure tin oxide. The high local pH created at −3 V by the $H_2$ evolution reaction rapidly etches and dissolves the native oxide from the surface since $SnO_2$ is unstable at pH ≥10.5 forming $Sn(OH)_{6,(aq)}^{2-}$[30,31]. The preferential dissolution of $SnO_2$ most likely happens at grain boundaries[27]. This is followed by fast nucleation and deposition of nanoparticles but nanoparticle growth is hindered by the high $H_2$ gas surface coverage produced at −3 V starving the sites from $Sn(OH)_{6,(aq)}^{2-}$ needed for further growth. Therefore, the diameter of nanoparticles produced at −3 V is only between 30 to 70 nm (Fig. 1a).

The XPS characterization of the three surfaces is shown in Fig. 1b. Tin oxide surface films are commonly composed of three tin species: Sn, SnO, and $SnO_2$, respectively. The relative contribution of each species was obtained by deconvoluting the XPS spectral peak associated with the Sn atom 3d5/2 orbital[32]. The blank polished surface is covered by a native passivating oxide layer that halts further atmospheric oxidation[33]. The surface with the highest proportion of $SnO_2$ is surprisingly the one pre-electroreduced at −1.25 V, a potential at which $SnO_2$ should be thermodynamically unstable in the $KHCO_3$ solution[5,34]. This unexpected increase in $SnO_2$ has been explained by the atmospheric oxidation of freshly reduced metallic Sn[35]. The pre-electroreduction at −1.25 V strips off part of the passivating layer, which partly uncovers the pure metallic tin grain, all the while leaving the more stable oxide on the surface. After drying, the freshly exposed tin metal reacts readily with atmospheric oxygen, oxidizing it to $SnO_2$[36]. This yields a final surface richer in $SnO_2$ than the unreduced native (blank) sample. Furthermore, through this process of electrochemical stripping of the oxide, followed by

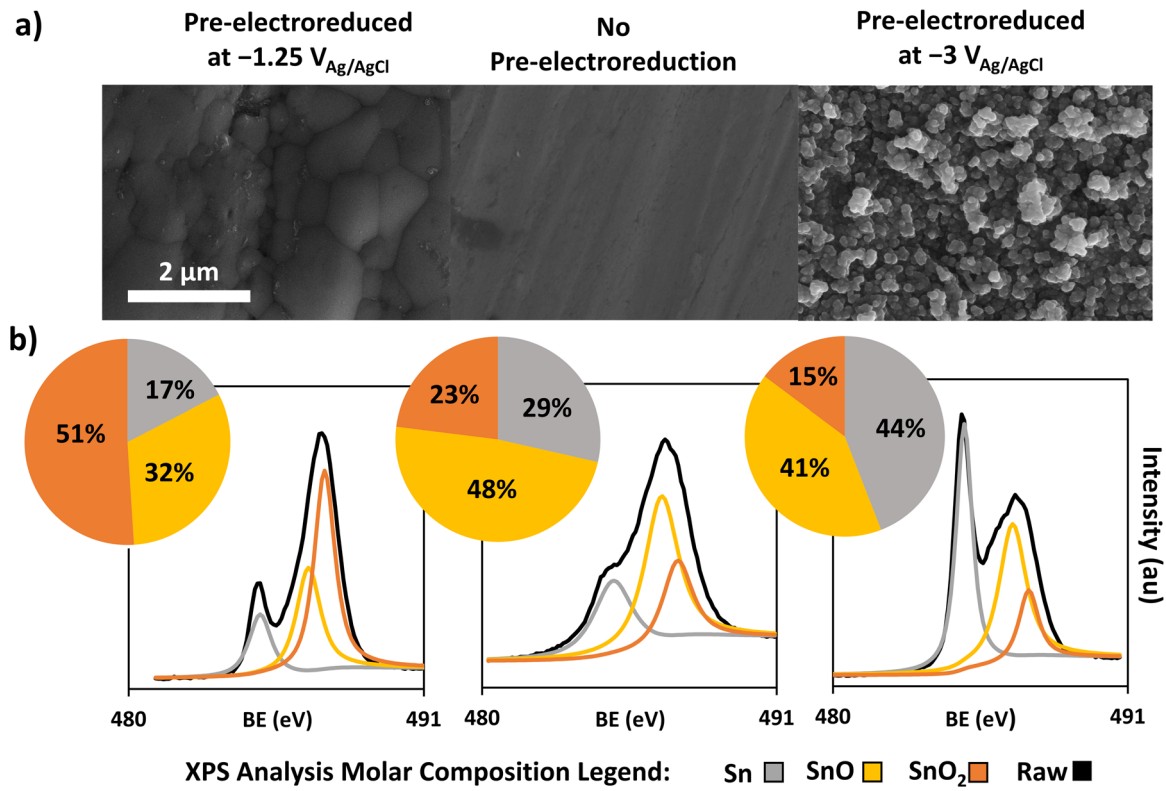

**Fig. 1 Sn/SnOₓ catalyst array characterization. a** SEM imaging, (**b**) XPS spectra and deconvolution at the Sn 3d5/2 orbital with surface molar compositions calculated based on peak surface areas.

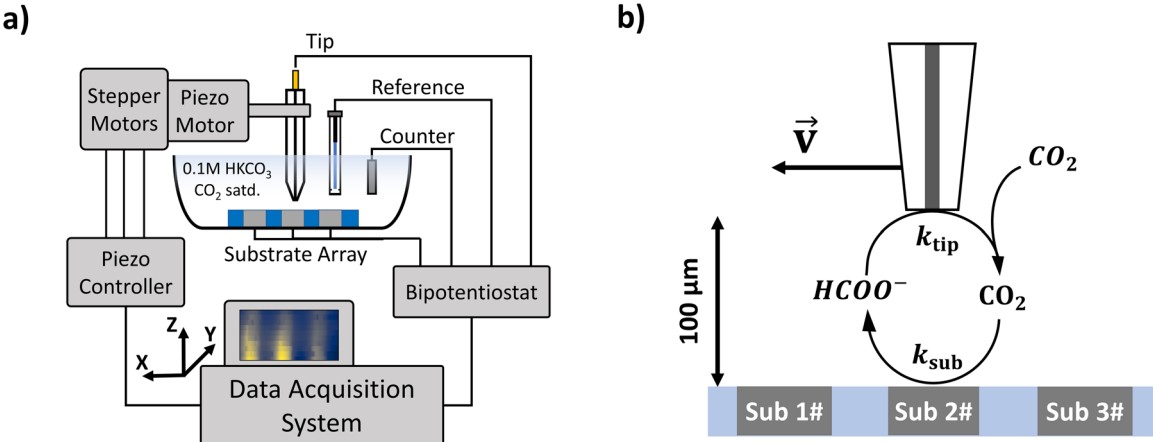

**Fig. 2 Schematic diagram of the SECM experiment for CO2RF with ultramicroelectrode detection. a** System diagram. (**b**) Schematic diagram of the substrate catalyst (Sn/SnOₓ) array for CO2RF with Pt tip scanning and in situ detection of formate.

atmospheric oxidation, it is likely that surface defects are also introduced that can also impact the electrocatalytic activity. The same phenomenon is at play for the substrate pre-electroreduced at −3 V, which has the lowest oxide fraction. In this case, the very negative electroreduction potential completely strips the surface of its oxides and the subsequent atmospheric oxidation would be unable to completely balance the loss of oxide incurred during the pre-treatment. This phenomenon yields a surface that still has a high proportion of oxides despite the strongly reducing condition of the pre-treatment.

**Characterization of the Pt ultramicroelectrode (SECM Tip) for probing CO2RF.** In this study, SECM is used in the substrate generation/tip collection mode (SG/TC) where both the tip and

substrate potential are independently controlled, and the tip current is recorded (Fig. 2a). The tip moves in an XY plane parallel to the plane of the substrate array, while electrochemically probing the chemical species in the diffusion shell created by the reaction at the substrate[37]. Ideally, dissolved $CO_2$ is reduced solely to formate and the resulting formate is oxidized at the tip (Fig. 2b). The ability for the Pt ultramicroelectrode (UME, 10 μm diameter) to act as a probe for formate detection in SECM was investigated. To evaluate the response of the Pt UME to formate in order to serve as a reference for the in situ detection, cyclic voltammograms (CV) were recorded with a scan rate of 1 V s⁻¹ in both $CO_2$ and $N_2$ saturated 0.1 M KHCO₃ with varying concentration of externally added potassium formate (Fig. 3a, b). The $N_2$ purged KHCO₃ solution represents model conditions, while

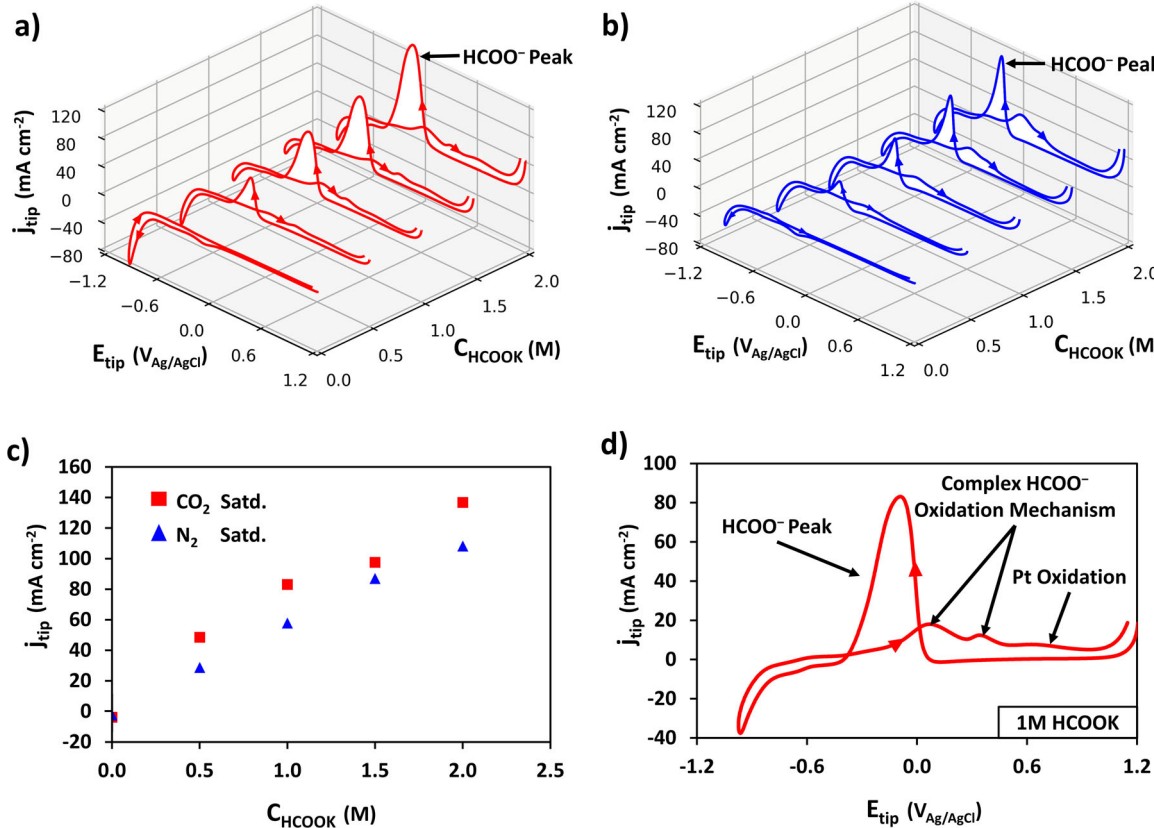

**Fig. 3 Platinum ultramicroelectrode (UME, 10 μm diameter) tip response for externally added potassium formate in 0.1 M KHCO₃ at 293 K. a, b** Cyclic voltammograms (50th cycle) with varying concentrations of potassium formate (between 0.5 to 2.0 M), (**a**) CO₂ saturated (pH 6.75) and (**b**) N₂ saturated (pH 8.75). **c** HCOO⁻ oxidation peak current density obtained in CO₂ saturated (red) and N₂ saturated (blue) electrolyte as a function of formate concentration. **d** Detail of a representative cyclic voltammogram on the Pt UME tip in CO₂ saturated (at atmospheric pressure) 1 M HCOOK solution. Scan rate: 1 V s⁻¹.

the $CO_2$ purged $KHCO_3$ solution represents the real conditions encountered during in-situ detection associated with the SECM scan. Comparison between the two conditions is necessary to assess the pH effect (6.75 vs. 8.75) on the formate CV. It must be noted that the pH can increase during the SECM scans of $CO_2$ electroreduction catalysts as a result of the competing $H_2$ evolution reaction on the catalyst substrate and $CO_2$ reduction to $HCOO^-$ according to the following reaction: $CO_2 + H_2O + 2e^- \rightarrow HCOO^- + OH^-$.

The CVs presented in Fig. 3a, b show clear electrochemical responses due to the presence of formate, similar to what has been reported for Pt macro-electrodes[38–40]. The most salient feature is the single sharp peak (i.e., $HCOO^-$ peak) on the cathodic scan direction at peak potentials between −0.1 and −0.3 V (dependent on the pH) (Fig. 3a, b). This peak has been associated in the literature with single step formate oxidation, forgoing the production of adsorbed CO intermediate[38–40]. As shown by Fig. 3c, in the $CO_2$ purged electrolyte, at lower pH, the $HCOO^-$ peak oxidation current density at the tip was slightly higher due to the pH effect on the $HCOO^-$ oxidation mechanism[40]. Importantly, in either $N_2$ or $CO_2$ purged conditions a correlation between the $HCOO^-$ peak current density and concentration was established (Fig. 3c).

Peaks in the anodic scan direction are associated with a more complex (two-step) formate oxidation mechanism with $CO_{ad}$ formation (at 0.05 V) and oxidation (at 0.34 V), as well as Pt oxidation[39] (Fig. 3d).

Next the Pt UME tip's ability for detection of formate generated in situ by $CO_2$ electroreduction on tin oxide catalysts was

investigated. CVs were recorded at the tip at 100 μm distance from the surface of the $Sn/SnO_x$ catalyst (Fig. 4a). The tip CVs were obtained at multiple points over the substrate, repeating the process for a series of decreasing substrate potentials (from −1.0 to −1.7 V). On $Sn/SnO_x$ catalysts, as a function of electrolyte composition and electrode potential, three species are expected to be produced: formate, CO and $H_2$[41]. The tip CVs in Fig. 4a clearly show in situ formate generation on native $Sn/SnO_x$ at substrate potentials starting at −1.2 V. In the cathodic scan direction, the $HCOO^-$ peak is representative for direct single-step formate oxidation to $CO_2$ and is similar to the response observed for externally added formate (compare Figs. 4a and 3a, b).

Analyzing now the peaks obtained in the anodic scan direction, an oxidation peak response is developing on the Pt tip at substrate potentials of −1.2 V and higher (in absolute value). This oxidation peak becomes a broad wave (extending between tip potentials of −0.6 V and 0.6 V) in the case of substrate potentials of −1.5 and −1.6 V, respectively (Fig. 4a). The broad oxidation wave is a feature of the electrocatalytic $CO_2$ reduction reaction (CO2RR), since Fig. 3a, b with externally added formate do not show the same broad oxidation wave on the anodic scan. The latter wave is due to a combination of formate oxidation plus interference from oxidation of the $H_2$ evolved (particularly at lower tip potentials starting at approximately −0.6 V) and contributions from the adsorbed CO peak at high tip potentials (between 0.3 and 0.6 V)[15,23,25]. At more negative catalyst substrate potentials $H_2$ production is favored over CO2RR, which explains the large anodic scan oxidation current on the Pt tip around −0.5 V. The reaction mechanism for $H_2$ oxidation on Pt has been thoroughly studied[42,43], but it can also influence the

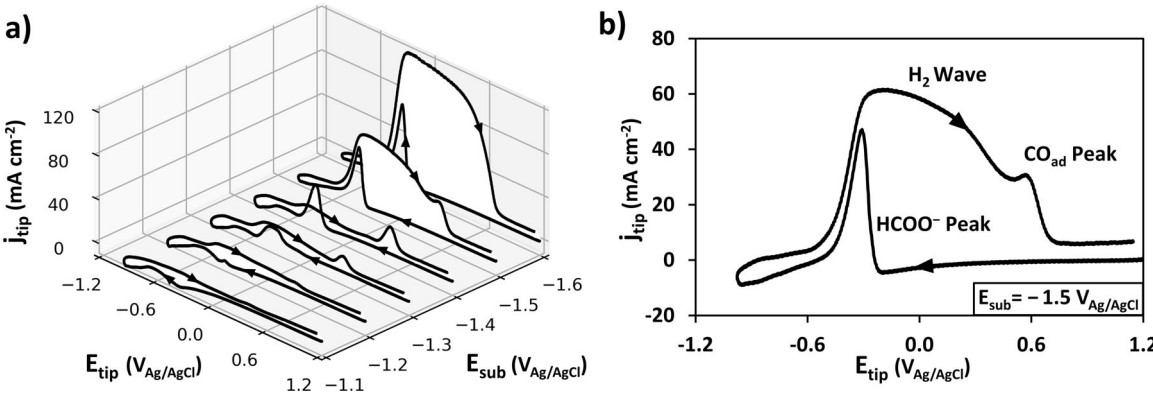

**Fig. 4 In situ detection of products generated by $CO_2$ electroreduction on $Sn/SnO_x$ catalyst. a** Pt UME tip CV response (scan rate $1 V s^{-1}$) for different catalyst substrate potentials ($-1.1$ to $-1.6 V_{Ag/AgCl}$), (**b**) Detailed Pt tip CV at a substrate potential of $-1.5 V_{Ag/AgCl}$. Electrolyte: $0.1 M$ $KHCO_3$ saturated with $CO_2$ at atmospheric pressure. 293 K. Catalyst: native $Sn/SnO_x$ (i.e., no pre-electroreduction, Fig. 1). The 1st CV cycles are shown.

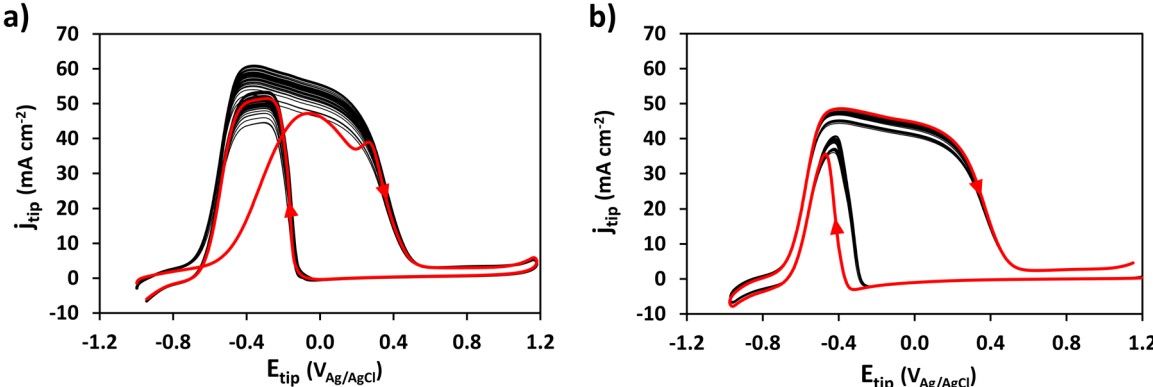

**Fig. 5 Effect of starting potential and repetitive cycling on the Pt UME tip CV for in-situ detection of products generated on the native (unreduced) $Sn/SnO_x$ catalyst.** Overlay of 50 CVs. **a** Starting potential: $-1.0 V_{Ag/AgCl}$, (**b**) Starting potential: $1.2 V_{Ag/AgCl}$. The 1st cycle is highlighted in red. Note: Before data acquisition for each cycle, the tip was held at the starting potential for 10 s to simulate the SECM scanning conditions. Electrolyte: $CO_2$ saturated (at atmospheric pressure) $0.1 M$ $KHCO_3$. Scan rate: $1 V s^{-1}$. 293 K. The tip-substrate distance: $100 \mu m$.

competitive adsorption and oxidation of formate and CO on Pt (Supplementary Figs. 3 and 4). However, in spite of these inherent complexities, sufficient information can be gained from the anodic and cathodic scan directions in the fast CV (Fig. 4b) to obtain a reasonable assessment of the product specific electrocatalytic activities.

At more positive substrate potentials ($-1.2 V$ and $-1.4 V$, Fig. 4a), CO is oxidized on the tip as a clearly distinguishable separate peak which is lumped into the broad wave at more negative substrate potentials ($-1.5 V$ and $-1.6 V$, Fig. 4a). The CO response can be due to two sources: oxidation of CO produced as intermediate of the formate partial oxidation on the Pt tip and CO produced by $CO_2$ electroreduction on $Sn/SnO_x$ (Supplementary Fig. 3). It is difficult to accurately and quantitatively separate these two contributions. Comparing Figs. 3a and 4a, the CO oxidation wave emerging from the formate pathway (Fig. 3a) is smaller even at high formate concentrations compared to the CO peak recorded during in-situ tip detection of $CO_2$ electroreduction products (e.g., at $-1.4 V$ substrate potential Fig. 4a). Therefore, it is proposed that the CO peak at the tip is mostly due to CO generated from CO2RR on $Sn/SnO_x$.

Given the presence of $CO_{ad}$, we investigated conditions that minimized the activity loss of the Pt UME tip for detection throughout the duration of the SECM scan[44]. Multiple sets of CVs were obtained on the tip placed close ($100 \mu m$) to an unreduced, native, $Sn/SnO_x$ catalyst held at a potential of $-1.5 V$. First, we tested the effect of starting potential. Two series of fifty

CVs were performed, one starting at $-1.0 V$ (Fig. 5a) and the other one starting at $1.2 V$. (Fig. 5b). Before each series of cycles, the tip was held for 10 s at the starting potential of the CVs to mimic the conditions between each data acquisition during SECM. The 10 s rest period at $1.2 V$ also serves a tip cleaning purpose. Thus, the stability of the tip response over fifty scans was much higher for a starting potential of $1.2 V$ indicating much less interference from $CO_{ad}$ poisoning. This observation was further substantiated in a series of extensive CV-SECM scans showing excellent reproducibility for three identical catalyst samples (Supplementary Figs. 5–7).

In terms of fast detection of CO2RR products at the Pt tip during SECM scanning over an array of catalysts, Figs. 3 and 4 highlight two approaches that could be employed. One approach, the simplest, is detection at a pre-selected constant tip potential giving a current response proportional to the total electrochemical activity of the catalyst(s) but without providing selective product detection. However, to obtain a measure of the specific catalytic activity for formate (i.e., CO2RF), the second approach of running fast scan CVs (e.g., at $1 V s^{-1}$) and extracting the tip current densities corresponding to the features discussed earlier (Fig. 4b) is more appropriate.

**SECM Screening of catalytic activity for CO2RR on an array of $Sn/SnO_x$ catalysts.** CV-SECM scans were performed on an array composed of three $Sn/SnO_x$ catalysts held at a constant potential

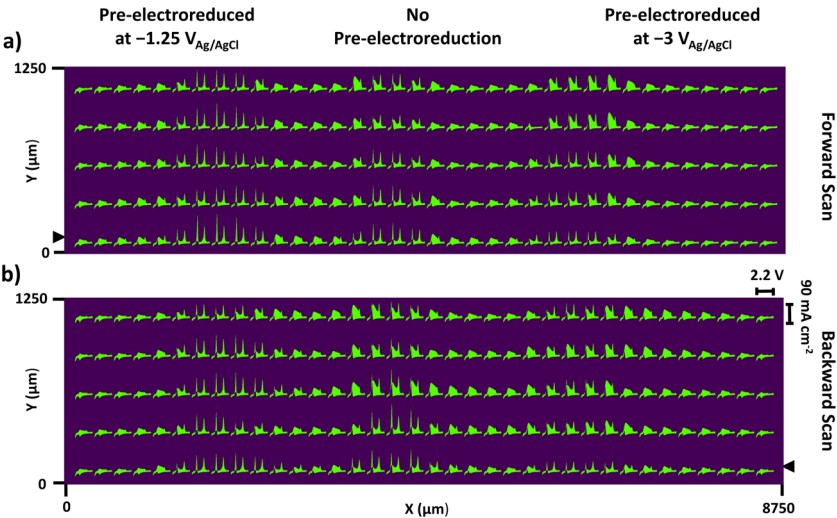

**Fig. 6 CV-SECM scans for a catalyst array with a substrate potential of −1.5 V$_{Ag/AgCl}$. a** Forward scan, (**b**) Backward scan. The black arrow indicates the starting scan position of the tip. CV: scan rate 1 V s$^{-1}$, potential range: 1.2 to −1.0 V$_{Ag/AgCl}$. Tip-substrate distance: 100 μm, tip scan rate 100 μm s$^{-1}$. Electrolyte: 0.1 M KHCO$_3$ saturated with CO$_2$ at atmospheric pressure, 293 K.

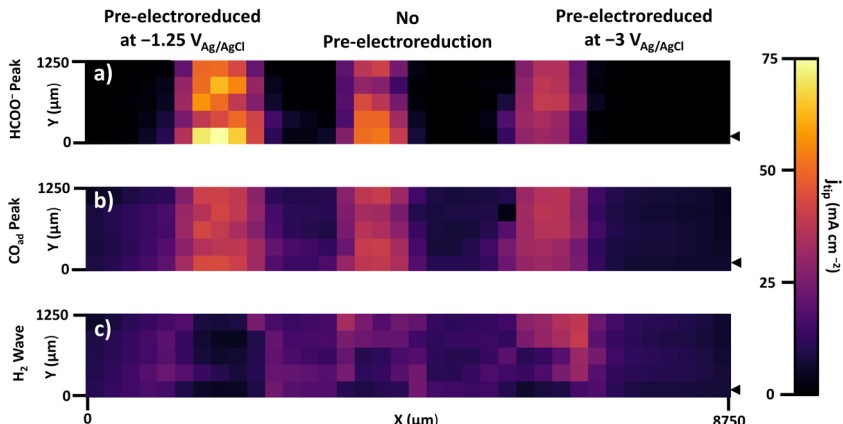

**Fig. 7 Pt UME tip current densities obtained from the backward CV-SECM scan (Fig. 6) and attributed to the three products of CO2RR. a** HCOO$^-$, (**b**) CO$_{ad}$, (**c**) H$_2$. Pixel size: 250 × 250 μm. The black arrow indicates the starting position of the tip. All conditions the same as in Fig. 6.

of −1.5 V (Fig. 6). Calibration scans were also carried out before and after the experiment to ensure that our data is free from artifact due to tilt and array fabrication defects (Supplementary Figs. 8 and 9).

Figure 6 shows the tip response is qualitatively distinct over the Sn/SnO$_x$ catalyst pre-electroreduced at −1.25 V, with HCOO$^-$ and CO peak dominating the CV. In comparison, the contribution from H$_2$ oxidation is more significant for the unreduced and the one pre-electroreduced at −3 V, with lower overall current density for the latter. For easier interpretation of results, the CV peak current density features attributed to formate, adsorbed CO, and H$_2$ oxidation, respectively, were extracted from each spatially resolved CV shown in Fig. 6 and were plotted for the backward CV-SECM scan in Fig. 7 as 2D images. The current densities plotted in Fig. 7 were extracted using a custom algorithm for CV processing and the following potentials were used: −0.456 V ±0.065 V from the cathodic scan direction (Fig. 7a), 0.453 V ±0.046 V (Fig. 7b) and −0.288 V ±0.034 V (Fig. 7c) from the anodic scan direction.

Starting with the direct formate oxidation peak (Fig. 7a), the catalyst with the highest activity for CO2RF is the one pre-electroreduced at −1.25 V, followed by the unreduced Sn/SnO$_x$. This is in accordance with the SnO$_2$ surface composition of the

catalysts (Fig. 1b): Sn/SnO$_x$ (−1.25 V) > Sn/SnO$_x$ (unreduced) > Sn/SnO$_x$ (−3 V). As was previously reported, the stripping and subsequent atmospheric oxidation of Sn increases its activity for CO2RF[10,36].

A few conclusions can be gleaned from the current density for the CO$_{ad}$ oxidation peak (Fig. 7b) and the H$_2$ oxidation wave (Fig. 7c). The CO$_{ad}$ tip response is comparable for each catalyst, with the substrate pre-electroreduced at −1.25 V exhibiting slightly higher CO oxidation current densities. More interestingly, the wave associated with H$_2$ oxidation is lacking for the sample pre-electroreduced at −1.25 V indicating virtually 100% Faradaic efficiency for CO$_2$ electroreduction to carbonaceous products (formate and CO). For the other two catalysts (pre-electroreduced at −3 V and the native surface, respectively), as the CV-SECM scan proceeds, the H$_2$ response increases while the formate response decreases a bit over time, indicating a slight degradation of the activity (Fig. 7c). This could be attributed to time dependent changes on the catalyst surface of the metastable Sn oxides. As the experiment proceeds with continuous exposure of the substrate at −1.5 V, in case of the high surface nanoparticle based catalyst prepared at −3 V with low initial SnO$_2$ content (Fig. 1), more of the metastable oxides are reduced to metallic Sn. Therefore, the activity toward H$_2$ evolution is increased on metallic Sn,

while supressing the CO2RF[36,45]. However, the catalyst prepared by pre-electroreduction at −1.25 V with the highest initial $SnO_2$ content (Fig. 1), is more stable under $CO_2$ reduction conditions and the activity toward formate generation is superior (Fig. 7). This ranking remained consistent throughout multiple trials. The catalyst produced by pre-electroreduction at −3 V was always less active for formate generation than the others in spite of its higher surface area. In future work, separate ex situ validation of these electrochemical results will be sought, by performing flow cell experiments with down selected individual catalysts (e.g., prepared by pre-electroreduction at −1.25 V) coupled with complete quantitative analysis of gaseous and liquid products.

Lastly, it is important to note that there are no significant interferences due to formate lateral diffusion during the time-frame of tip scanning from one catalyst to the next, as shown by the clearly delineated inert zone (i.e., occupied by the resin) separating the samples with virtually negligible tip current density (Fig. 7a). Thus, the latter phenomenon, that could also be referred to as catalyst 'cross-talk', plays no role with optimized tip speed, sample size and fast CV scan detection for formate detection. The lateral diffusional isolation is further demonstrated in Supplementary Figs. 5 and 6.

## Conclusion

We investigated the potential of SECM for electrocatalytic activity screening of $Sn/SnO_x$ based catalyst arrays for CO2RF. The latter reaction is of great interest for value added conversion of $CO_2$. The catalysts were prepared by electroreduction of mirror polished $Sn/SnO_x$ surfaces. We demonstrated that with proper characterization and calibration, simultaneous SECM scans over an array composed of three $Sn/SnO_x$ catalysts having different compositions (i.e., Sn, SnO, and $SnO_2$ ratio) and morphologies (i.e., ranging from smooth surface to spherical nanoparticle aggregates) generated reliable electrocatalytic activity data as a function of the applied substrate potential (between −1.2 and −1.6 V vs. Ag/AgCl). The electrocatalytic activity was quantified by in situ detection of the products ($HCOO^-$, CO and $H_2$) at a Pt ultramicroelectrode (i.e., scanning tip) subjected to fast ($1 V s^{-1}$) CV sweeps. The trend of relative electrocatalytic activities for formate production follows the catalysts' $SnO_2$ surface mole fraction: $Sn/SnO_x$ (−1.25 V) > $Sn/SnO_x$ (unreduced) > $Sn/SnO_x$ (−3.0 V). This ranking remained consistent throughout multiple trials. The catalyst produced by pre-electroreduction at −3 V with only 15% initial $SnO_2$ mole fraction was less active than the others in spite of its higher surface area, whereas the catalyst produced by pre-electroreduction at −1.25 V with 51% initial $SnO_2$ mole fraction, showed the highest activity and stability for formate and CO production.

Building on the principles outlined here, in future work the SECM technique could be extended to the investigation of larger catalyst arrays and matrices. Such an experimental technique, combined with artificial intelligence and machine-learning based processing of large data sets (i.e., high-throughput experimental screening), could greatly accelerate the catalyst discovery process for $CO_2$ electroreduction generating a range of valuable products such as formate, CO, hydrocarbons, alcohols. Furthermore, future work should also focus on the combination of CV-SECM scans with localized surface analysis techniques such as XPS, to map local hotspots and compositional inhomogeneities on catalyst surfaces.

## Methods

**Chemicals and materials**. Potassium chloride (99.0%), potassium bicarbonate (analytical quality), hydrochloric acid 36.5% V/V (ACS grade) were purchased from VWR analytical. Potassium hexacyanoferrate(II) trihydrate (98.5%), potassium formate (99%), and ferrocene methanol (97%) were purchased from Sigma Aldrich. Tin ingot (99.99%) was purchased from Alfa Aesar. Nitrogen gas (99.999%) and carbon dioxide (99.99%) gas cylinders were purchased from Praxair. All aqueous solutions were prepared using deionized water with a resistivity of $16.5 M\Omega cm^{-1}$.

**Substrate catalyst array preparation**. The catalyst array was created by embedding three individually wired 10 mm × 10 mm × 1 mm Sn foils (Puratronic, 99.9985%) in epoxy resin (System Three Cold Cure). The resulting array was polished using a series of silicon carbide metallurgical polishing paper (Allied High Tech Product) with standard grit size of 600, 800, and 1200, with tap water as lubricating fluid. The resulting mirror finished $Sn/SnO_x$ surface was the starting material for our catalyst samples. Subsequently, the entire array was subjected to a pre-electroreduction treatment, with each individual substrate being submitted to one of three pre-electroreduction treatments in $N_2$ saturated 0.1 M $KHCO_3$ solution. The three pre-electroreduction treatments were as follows: (i) reduction at −1.25 V for 30 min., (ii) reduction at −3 V for 30 min., and (iii) no pre-electroreduction (blank, native surface). The pre-electroreduction pre-treatment was ended by prompt removal of the catalyst array from the solution and washing with DI water. The reaction space over each substrate was segregated from the other by plastic microscope cover slips to prevent drift of the reduction product from one substrate to the other. There were approximately four hours between the end of the substrate array preparation and the start of the SECM experiment.

**Instrumentation**. SECM data was taken using a combination Heka ring/disk potentiostat PG 340 and Heka Elproscan controller ESC 3. SEM images were taken on a ZEISS FE SEM ULTRA 55 using an acceleration voltage of 10 kV. XPS measurements were performed with a Kratos Analytical Axis Ultra DLD. pH measurements were performed using an Oakton pH 110 series pH meter.

**SECM tip preparation procedure**. The tip used in this work was a 10 μm diameter platinum ultramicroelectrode (Sensolytics GmbH.) with an RG value around 25. The RG value refers to the ratio of the radius of the tip insulating sheet over the radius of the active area. The RG value of the tip was determined with an optical microscope (Olympus MG), imaging the bottom of the tip, where the Pt electrode is exposed. The tip was coarsely polished using a BV-10 Micropipette beveller (Sutter Instrument) and a coarse abrasive plate fine. This was followed by manual polishing with 1 μm monocrystalline diamond water based polishing suspension (Allied High Tech Product) on a 0.05 μm polishing cloth (Leco Imperial polishing cloth). The measured diameter of the tip was an average of 9.67 μm with a standard deviation of 0.86 μm. The true electroactive surface area of the tip was determined by the limiting current density method[46]. The testing solution was 0.1 M KCl with 30 mM of potassium hexacyanoferrate(II) trihydrate. Cyclic voltammograms obtained at 50 mV s$^{-1}$ were recorded between −0.2 to 0.8 V vs. Ag/AgCl to determine the limiting current density. All current densities are reported with respect to the measured electroactive area of the tip which was very similar to the calculated geometric electroactive area.

**SECM approach curve and tilt compensation**. The SECM scanning of the substrate composed of the $Sn/SnO_x$ catalyst array started with an approach curve performed using the $O_2$ dissolved in an air saturated 0.1 M KCl solution. Before the experiment, the cell was assembled and connected to a bipotentiostat. The approach curve consisted of holding the Pt ultramicroelectrode tip at a constant $XY$ position and slowly decreasing the Z position, all the while holding the tip at a constant potential where the oxygen reduction reaction takes place under diffusion-controlled conditions (−0.75 V vs. Ag/AgCl). Note: all potentials from here on are referenced against Ag/AgCl, KCl saturated. Initially, the tip was outside of the cell and the substrate was inside the cell. The electrolyte solution was poured inside the cell slowly. Once the solution contacted the substrate and the reference electrode, the remaining solution was poured very quickly to also contact the counter electrode (Pt wire, 1 mm diameter, 40 mm long). The tip was then lowered into the bulk solution far from the substrate array surface. A CV at the tip was obtained (scan rate: 50 mV s$^{-1}$, scan range: 0.0 to −1.0 V for 3 cycles). This CV was used to determine the tip potential at which the dissolved $O_2$ molecules are reduced at the tip. This CV also helps to detect any problems with the tip, in which case the tip would be replaced.

Factors influencing the tip approach curve include how long the solution was sitting in the cell and how close to the surface the tip was. The approach curve used the negative feedback mode of the SECM. By lowering the tip to closely approach the resin part of the array, the absolute current registered at the tip at a constant potential of −0.75 $V_{Ag/AgCl}$ decreased monotonically until it reached an inflection point. This inflection point indicates the position at which the tip hits the resin's surface. At this position, the resin's surface obstructs the diffusion of dissolved $O_2$ from the bulk solution to the tip electroactive surface, limiting the absolute current at the tip. An approach curve was done at each corner of the scan area (Scan area equal to: 1000 × 2250 μm), to determine the tilt of the array surface with respect to the SECM apparatus. Once the scan area tilt is calculated, it can be compensated by the piezo-electric actuator of the SECM. This allows the tip-substrate distance to be

kept constant throughout the scan despite slight misalignment between the SECM tip and the substrate sample.

A constant potential SECM scan was also performed over the substrate array in the approach curve solution (air saturated 0.1 M KCl), in order to detect any anomaly (e.g., poor substrate conductivity, gross surface deformation, poor to tilt compensation) that would taint the experimental result acquired subsequently. During this scan substrate potential remained −1.0 V but the tip potential was lowered to −1.0 V to better reduce $O_2$ at condition near the catalyst. This SECM scan was also used to precisely determine the position in the XY plane of the substrate in the array. As opposed to the other constant potential SECM scan in this work, this scan was performed using the redox competition (RC) mode[47,48], where both the substrate and tip electrode competed for the dissolved oxygen in the electrolyte.

**Preparation of the system for the SECM experiments**. Before the actual SECM scans are started, the solution from the $O_2$ reduction approach curve must be removed from the cell. The tip was moved up 2000 μm from the surface, the Pt tip potential was changed to −0.5 V and the 0.1 M KCl solution was pumped out of the cell through a previously installed Pasteur pipette inside the cell. The solution level in the cell was kept such as to prevent breaking the electrical circuit between the bi-potentiostat and both working electrodes (i.e., tip and substrate). The new electrolyte solution, (0.1 M KHCO₃) was poured gently inside the partially empty cell to top off the solution level. $CO_2$ was bubbled inside the cell through another Pasteur pipette to agitate the solution. After 5 min. the cell was again partially emptied and topped-up again with the new electrolyte solution. In total the cell was partially emptied and refilled four times. After this, the substrate potential was changed from −1.0 V to −1.5 V to pre-emptively reduce potential unstable tin oxide species that might dissolve and reprecipitate on the tip during the SECM scan. After waiting 20 min., the tip was brought back at 100 μm from the surface. Before the CV-SECM scan was performed, a constant potential SECM scan is performed with a tip potential of −0.5 V. The scan lasted 35 min. Results of these constant tip potential SECM scans are not presented in this work since the fast CV tip detection was preferred for selective product characterization (described below).

**CV-SECM scans**. In this case, the SG-TC mode of SECM was used. The CV-SECM scan was performed after a constant potential SECM (described above). The substrate potential was held at −1.5 V. The tip CV sweep consisted of recording a single CV (between 1.2 to −1.0 V, with a scan rate of 1 V s$^{−1}$) at each 250 μm in the forward X direction at a constant Y position. Between each acquisition point the tip was at a constant potential of 1.2 V for cleaning. This was followed by a scan along the same Y coordinate, but in the backward X direction. Once both scans were acquired, the tip was moved 250 μm in the Y positive direction. Once at the new position, the single line scan procedure was repeated until ten backward and ten forward X direction scans were completed. The total scan area was 8750 μm by 2250 μm, which yielded 36 acquisition points per line scan. A complete scan lasted for 2.5 h.

**SECM redox mediator feedback scan**. After the SECM experiment, the tip was removed from the cell and the substrate potential was held at −1.5 V. The electrolyte solution was removed from the cell and topped off with DI water without breaking the circuit. This was repeated three times over the span of 20 min. while N₂ gas was bubbled through the cell, mixing the solution. The last top-up involved a N₂ saturated 0.1 M KCl, 3 mM ferrocene methanol solution instead of DI water. A freshly polished tip of known electroactive diameter was inserted in the cell, followed by a tip CV scan (−0.2 to 0.6 V, 50 mV s$^{−1}$, 3 cycles). Limiting current was extracted from the CV and used to determine the concentration of ferrocene methanol in the cell[46]. An approach curve was performed ($E_{tip} = 0.3$ V) toward the resin surface of the array to determine the new position of the substrate with respect to the tip. Using the new surface position and previously determined tilt, an SECM scan was performed over the array ($E_{tip} = 0.3$ V, $E_{sub} = −1.5$ V). This scan was used to detect any distortion in tip-substrate distance, or any other artifact that could have developed during the experiment.

**XPS analysis sample preparation**. Samples for XPS analysis were prepared by polishing Sn (Alfa Aesar 99.99%) pieces of 10 × 10 × 1 cm which were inserted into a specialized electrochemical cell. The specialized cell assembly enables connection to the potentiostat while preventing contamination of the sample. The substrate electroreduction was performed with the same procedure as for the substrate arrays (presented before). At the end of the electroreduction procedure, the sample was removed from the solution and washed with DI water. The substrates were stored individually in a glass Petri dish, with their backside glued to the back of the Petri dish with double sided carbon tape. XPS analysis was performed about a week after the substrate pre-electroreduction.

## Data availability
The datasets generated and analyzed during this study are available from the corresponding authors based on request.

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

# ARTICLE

26. Voort G. F. Vander. Metallography and microstructures of tin and tin alloys. In *Metallography and Microstructures*. Vol. 9, 889–898 (ASM International, 2004).

27. Padrós, A. P. Anodic tin oxide films: fundamentals and applications. Ph.D. Thesis, University of Barcelona, Spain (2015).

28. Huang, C. A., Li, K. C., Tu, G. C. & Wang, W. S. The electrochemical behavior of tin-doped indium oxide during reduction in 0.3 M hydrochloric acid. *Electrochim. Acta* **48**, 3599–3605 (2003).

29. Liu, L., Yellinek, S., Valdinger, I., Donval, A. & Mandler, D. Important implications of the electrochemical reduction of ITO. *Electrochim. Acta* **176**, 1374–1381 (2015).

30. Rai, D., Yui, M., Schaef, H. T. & Kitamura, A. Thermodynamic model for $SnO_2(cr)$ and $SnO_2(am)$ solubility in the aqueous $Na^+$–$H^+$–$OH^-$–$Cl^-$–$H_2O$ System. *J. Solut. Chem.* **40**, 1155–1172 (2011).

31. Schweitzer, G. K. & Lester L. Pesterfiel. *The Aqueous Chemistry of the Elements*. (Oxford University Press, 2010).

32. Jie, L. & Chao, X. XPS examination of tin oxide on float glass surface. *J. Non Cryst. Solids* **119**, 37–40 (1990).

33. Giannetti, B. F., Sumodjo, P. T. A., Rabockai, T., Souza, A. M. & Barboza, J. Electrochemical dissolution and passivation of tin in citric acid solution using electron microscopy techniques. *Electrochim. Acta* **37**, 143–148 (1992).

34. Dutta, A., Kuzume, A., Rahaman, M., Vesztergom, S. & Broekmann, P. Monitoring the chemical state of catalysts for $CO_2$ electroreduction: an in operando study. *ACS Catal.* **5**, 7498–7502 (2015).

35. Khatavkar, S. N., Ukale, D. U. & Haram, S. K. Development of self-supported 3D microporous solder alloy electrodes for scalable $CO_2$ electroreduction to formate. *N. J. Chem.* **43**, 6587–6596 (2019).

36. Zhang, R., Lv, W. & Lei, L. Role of the oxide layer on Sn electrode in electrochemical reduction of $CO_2$ to formate. *Appl. Surf. Sci.* **356**, 24–29 (2015).

37. Bard, A. J. & Mirkin, M. V. *Scanning Electrochemical Microsocopy*. 538–542 (CRC Press, 2012).

38. Sun, S. G., Clavilier, J. & Bewick, A. The mechanism of electrocatalytic oxidation of formic acid on Pt (100) and Pt (111) in sulphuric acid solution: an EMIRS study. *J. Electroanal. Chem. Interfacial Electrochem.* **240**, 147–159 (1988).

39. John, J., Wang, H., Rus, E. D. & Abruña, H. D. Mechanistic studies of formate oxidation on platinum in alkaline medium. *J. Phys. Chem. C.* **116**, 5810–5820 (2012).

40. Brimaud, S., Solla-Gullón, J., Weber, I., Feliu, J. M. & Behm, R. J. Formic acid electrooxidation on noble-metal electrodes: role and mechanistic implications of pH, surface structure, and anion adsorption. *ChemElectroChem* **1**, 1075–1083 (2014).

41. Irtem, E. et al. Low-energy formate production from $CO_2$ electroreduction using electrodeposited tin on GDE. *J. Mater. Chem. A* **4**, 13582–13588 (2016).

42. Zheng, J., Sheng, W., Zhuang, Z., Xu, B. & Yan, Y. Universal dependence of hydrogen oxidation and evolution reaction activity of platinum-group metals on pH and hydrogen binding energy. *Sci. Adv.* **2**, 1–9 (2016).

43. Durst, J. et al. New insights into the electrochemical hydrogen oxidation and evolution reaction mechanism. *Energy Environ. Sci.* **7**, 2255–2260 (2014).

44. Jambunathan, K., Shah, B. C., Hudson, J. L. & Hillier, A. C. Scanning electrochemical microscopy of hydrogen electro-oxidation. Rate constant measurements and carbon monoxide poisoning on platinum. *J. Electroanal. Chem.* **500**, 279–289 (2001).

45. Pander, J. E., Baruch, M. F. & Bocarsly, A. B. Probing the mechanism of aqueous $CO_2$ reduction on post-transition-metal electrodes using ATR-IR spectroelectrochemistry. *ACS Catal.* **6**, 7824–7833 (2016).

46. Baur, J. E. & Wightman, R. M. Diffusion coefficients determined with microelectrodes. *J. Electroanal. Chem. Interfacial Electrochem.* **305**, 73–81 (1991).

47. Eckhard, K. & Schuhmann, W. Localised visualisation of $O_2$ consumption and $H_2O_2$ formation by means of SECM for the characterisation of fuel cell catalyst activity. *Electrochim. Acta* **53**, 1164–1169 (2007).

48. Okunola, A. O. et al. Visualization of local electrocatalytic activity of metalloporphyrins towards oxygen reduction by means of redox competition scanning electrochemical microscopy (RC-SECM). *Electrochim. Acta* **54**, 4971–4978 (2009).

## Acknowledgements

The financial support from NSERC Canada through the Discovery Grant program (awarded to E.G. and separately to E.A.) is gratefully acknowledged. The Mitacs GlobaLink grant awarded to F.M. for a research visit at CNRS Paris is greatly appreciated. The authors greatly appreciated the thorough and insightful reviews of the manuscript.

## Author contributions

E.G., E.A., C.S.S., and P.H.B designed the study. F.M. prepared the catalysts and performed all the experiments under the guidance of P.H. F.M. and E.G. wrote the paper with input from C.S. and E.A. All authors discussed the results and contributed to the manuscript.

## Competing interests

The authors declare no competing interests.
