## [Peer Review File · Communications Chemistry]

Reviewers' comments:

Reviewer #1 (Remarks to the Author):

Comments on COMMSCHEM-20-0192-T

The present manuscript reported the characterization of CO₂RF catalyst arrays by SECM. Revealing more spatially resolved properties of catalysts is of great interest in improving catalyst performance. Though some results might be helpful, overall the organization of the manuscript should be further improved. In addition, the novelty of the manuscript is not well justified, especially with many risky statements. A major revision is recommended before its publication:

1. Title is somewhat misleading. The manuscript seems to be more of new preparation method, instead of a SECM development. Authors are suggested to be clarify on this to avoid possible confusion.
2. A major drawback of SnO₂ was mentioned in the first paragraph of introduction. However, authors are suggested to further elaborate how the implementation of SECM helped to solve the problem.
3. Authors claimed that the novelty of this manuscript is that they used SECM for screening of catalyst arrays and previous SECM studies focused on signal catalyst characterization. It seems that this is an improvement for sample preparation, not an innovation relevant to SECM technique itself.
4. Line 86-87, it is hard to tell just from the figure1 & S1 that why did authors choose the sample treated with -3V not -3.5V and sample treated with -1.25V not -1.5V (or -2V). The sample images of -3V and -3.5 V seem to be similar to each other.
5. Line 140-142, when characterizing the Pt Microelectrode, authors are suggested to explain why they used and compared CO₂ and N₂ saturated solution.
6. The experiment design for "Pt ME tip's ability for SG/TC detection" may be problematic. Line 143, in SG/TC mode, the formate should be in situ generated near the substrate. However, the formate was externally added into the solution.
7. Line 145, "The 100 μm was chosen as the tip-substrate distance to mitigate artifacts due surface roughness." What is the sample surface roughness value?
8. Since authors used different scale bar and color, it's difficult to compare the images and results in Figure 5 with Figure 4, especially Figure 4A.

Reviewer #2 (Remarks to the Author):

In this paper, Mayer et al reported the development of scanning electrochemical microscopy for high-throughput screening of CO₂ reduction electrocatalysts. Although similar ideas have been reported previously by several research groups, such as Bard, Unwin, White, this paper contains some information that is useful for the researchers in the related fields and the manuscript is also well written although some improvement in clarity is still needed as stated below. Therefore, in my view, it is acceptable for publication after major revision taking into the account the following comments:

- The authors use the term "2D scanning electrochemical microscopy" throughout (e.g., line 19 in the abstract). It wasn't clear to me what the 2D refers to? The title of the paper is also confusing. What does high throughput electrocatalysis mean?
- In the abstract (and conclusions), the authors allude to "product-selective detection". One well-known limitation of amperometric detection (such as SG/TC SECM) is the lack of selectivity (also seen in the paper, as the SECM tip signal arises from both formate and H₂ oxidation). There are other product produced at the substrate such as CO. Is the tip potential of -0.5 V for SECM 2D scans selective to the formate? This needs to be rationalised further.

- Another known limitation of using a solid electrode probe to monitor an electrocatalytic process is that the SECM tip can be prone to “fouling”, meaning that the signal at the tip changes with time. This is shown in Figure S4, Figure S5, and is also seen Figure S6 (the response changes with time/cycling). This is likely to become more problematic with more complicated product detection (see “product-selective detection” comment above). How can such an effect be prevented/accommodated for?
- In Fig. 3C, why is there a difference between CO₂ and N₂ atmosphere? Furthermore, why is there a non-zero intercept?
- SECM is nontrivial to be used as a screening technique (i.e., low throughput per sample due to long experimental time) and it is more suitable to study kinetics and spatial heterogeneity of the catalyst in the fundamental aspect. The authors refer to the presented method as “high-throughput” throughout (e.g., Line 193). When reading through the Experimental section, it seems that a full set of experiments would take a few hours to carry out (e.g., the fast-scan CV scan takes 2.5 hours alone), so my question is: is this truly a high-throughput method? It seems like the equivalent set of experiments could be carried out a rotating ring-disk electrode (RRDE) on a similar timescale. The author should revisit the introduction and emphasise more specifically why the SECM is important for this work.
- On Line 200: “free from artifact due to tilt, roughness and resistivity of the substrate array”. Surface roughness can cause artifacts for most scanning probe methods and SECM is no exception. As real electrocatalysts tend to possess a degree of roughness (noted in Figs. S1 and S2), is this technique truly applicable with “non-model” substrates, as claimed in the introduction (lines 68 – 75)? Additionally, how can the authors be confident that an increase in current detected at the SECM is due to an increase in activity rather than just an increase in catalytic surface area?
- In Fig. 4A, what is the y-contrast in the pre-reduced at -1.25 V sample (left-most plot)? It looks like the top of the sample possesses a different activity to the bottom. Is this an artifact caused by tip/substrate fouling?
- On line 219, the authors mention “catalyst ‘cross-talk’, plays no role...”, but in Fig. 4A, S7 and S8, the three catalysts do not appear to be diffusionally isolated, apparent from the non-zero current detected between the Sn/SnO_x strips.
- Further to the points above, how feasible would it be to expand this method to a larger array of catalysts? My feeling is that artifacts from sample tilt, tip-fouling and diffusional cross-talk (see comments above) would ultimately limit the applicability of this technique.
- On lines 220 – 223 “The 2D spatial variation within a sample can be attributed to a large degree to the non-homogeneous catalyst surface in terms of the distribution of SnO_x and metallic Sn.” This is potentially a very important observation, as it indicates that the catalysts are not homogeneously active. Is it possible to couple the SECM scanning with (ex situ) co-located compositional/structural analysis (e.g, XPS mapping) to understand the nature of the catalytic hotspots? Or is it possible that the apparently “high-activity” areas are simply more rough/porous, giving rise to a higher catalytic surface area?
- Reading through the Experimental section, the catalyst undergoes prolonged exposure to aerated 0.1 M KCl prior to scanning. Oxygen reduction is also performed on the catalyst surface (e.g., Fig. S7) prior to CO₂ electroreduction screening. How do these protocols affect the surface structure/composition of the electrocatalysts? Additionally, could surface contamination with chloride and/or oxy/hydroxy species be a contributor to the observed trends? (i.e., could the -3 V sample be more susceptible to contamination/fouling, leading to apparently lower activity?)
- Main results in Figure 4 and Figure 5: The author needs to be more clear about the results from “SECM 2D scans with constant tip potential”. As the author addressed already, the Pt is contaminated with CO and SnO₂ and this significantly changes electrochemical activity of the probe (figure S4 and S5). As shown in the image in the Figure 4, the current density is two orders of magnitude lower than what it should be when Figure 4A is compared to Figure 5B (as well as Peak 1

in Figure 3D). The author needs to validate what it is measures in the “SECM 2D scan with constant tip potential” and if it is directly relevant to formate detection.

- Minor comments: A lot of figure captions are lack of details of the figures in both main text and the supporting information. The author needs to ensure to provide all details of the figure to the readers.

Rebuttal Letter:

The authors greatly appreciated the thorough and insightful reviews of the manuscript. All the suggestions and recommendations were taken into account and, as detailed in the point-by-point answers below, appropriate changes and revisions were made, which are also highlighted in yellow in the revised manuscript.

Reviewer #1:

Though some results might be helpful, overall the organization of the manuscript should be further improved. In addition, the novelty of the manuscript is not well justified, especially with many risky statements. A major revision is recommended before its publication:

1. Title is somewhat misleading. The manuscript seems to be more of new preparation method, instead of a SECM development. Authors are suggested to be clarify on this to avoid possible confusion.

The title was revised and the new title is: *Application of Scanning Electrochemical Microscopy for Electrocatalytic Activity and Product Selective Screening of CO₂ Reduction Catalyst Arrays*

The manuscript has a number of novelties, in addition to the preparation of the catalysts and catalyst array. Thus, it is important to note that this is the first report where SECM is used for simultaneous electrocatalytic activity screening of three catalysts arranged in an array. Besides the preparation of the array, in order to successfully accomplish the afore-mentioned objective, we performed detailed tip characterization studies (e.g., by fast cyclic voltammetry) to detect all three products of CO₂ reduction and HER: HCOO⁻, CO and H₂. In the revised manuscript we provide further details and results regarding fast CV-SECM scans with selective detection of the products (please see Figs. 4-6 in the revised MS). In this study we lay some of the groundwork for the application of SECM for screening catalyst arrays and matrices, which could lead to high-throughput screening and acceleration of the catalyst discovery process. Furthermore, another novelty is the finding that the Sn/SnO_x catalyst produced by the pre-electroreduction method at -1.25 V vs Ag/AgCl displays remarkable intrinsic activity and durability.

2. A major drawback of SnO₂ was mentioned in the first paragraph of introduction. However, authors are suggested to further elaborate how the implementation of SECM helped to solve the problem.

The sentence mentioned is: “One of the most promising catalyst for this process is SnO₂, which has shown high selectivity and low overpotential for CO₂RR^{3,4}. However, the electrochemical reduction of SnO₂ to the less active metallic form at potentials relevant for CO₂RR^{5,6}, is a major drawback.”

While the major goal of our paper is not this issue, we found that among the three catalysts we investigated, the Sn/SnO_x produced by the pre-electroreduction method at -1.25 V vs Ag/AgCl had the highest activity for formate generation. This catalyst had the highest initial content of SnO₂ on the surface and it appeared to be more stable compared to the other formulations. This finding necessitates further detailed catalysis studies, which were beyond the objective here.

3. Authors claimed that the novelty of this manuscript is that they used SECM for screening of catalyst arrays and previous SECM studies focused on signal catalyst characterization. It seems that this is an improvement for sample preparation, not an innovation relevant to SECM technique itself.

As we also mentioned in our answer to question 1, one novelty of this manuscript is the use of SECM to selectively detect and quantify *in situ* the products from CO₂RR by fast (1 V s⁻¹) CV at the Pt ultramicroelectrode. In particular, we address the formate generation activity and we link those results to the presence of specific oxides on the Sn surface quantified by XPS.

In the revised manuscript we emphasize more forcefully these aspects, including the Abstract and Conclusion.

4. Line 86-87, it is hard to tell just from the figure 1 & S1 that why did authors choose the sample treated with -3V not -3.5V and sample treated with -1.25V not -1.5V (or -2V). The sample images of -3V and -3.5 V seem to be similar to each other.

The reviewer is right. Some pre-treatment potentials provided similar morphology samples. This is the case for samples treated at -1.0, -1.25, -1.5 and -2 V as well as samples treated at -3 and -3.5 V. For this reason, we only selected for comparison the two main types of morphology obtained together with the native Sn/SnO_x sample.

In the revised manuscript on p.4, we specified these pre-treatment potential dependencies and made clear our choice of samples.

5. Line 140-142, when characterizing the Pt Microelectrode, authors are suggested to explain why they used and compared CO₂ and N₂ saturated solution.

The N₂ purged KHCO₃ solution represents model conditions, while the CO₂ purged KHCO₃ solution represents the real conditions encountered during *in situ* detection associated with the SECM scan. Comparison between the two conditions is necessary to assess the effect of pH (6.75 vs. 8.75) on the formate CV, which might also change during the SECM scans due to the competing H₂ evolution reaction and CO₂ reduction as follows: $\text{CO}_2 + \text{H}_2\text{O} + 2\text{e}^- \rightarrow \text{HCOO}^- + \text{OH}^-$. As shown by Fig. 3C, in the CO₂ purged electrolyte at lower pH the formate oxidation current density at the tip was slightly higher, however, a linear correlation between oxidation current density and formate concentration could be established in both cases.

This explanation was added to the revised manuscript on p: 5 to explain the role of the two conditions.

6. The experiment design for "Pt ME tip's ability for SG/TC detection" may be problematic. Line 143, in SG/TC mode, the formate should be in situ generated near the substrate. However, the formate was externally added into the solution.

The paragraph devoted to the Pt UME tip characterization for probing CO₂RR in Figures 3A, 3B, 3C, 3D, with externally added potassium formate has the goal to establish a reference in terms of formate CVs and peak current densities for the subsequent *in situ* detection of

formate generated by CO₂ reduction on Sn/SnO_x, which is discussed at length in the subsequent paragraphs related to Figs. 4 to 7.

In the revised manuscript this was stated more clearly and the paragraph related to the tip characterization with externally added formate related to Fig.3 was revised.

7. Line 145, “The 100 μm was chosen as the tip-substrate distance to mitigate artifacts due surface roughness.” What is the sample surface roughness value?

We did not measure the surface roughness but based on empirical observations we never had any issue when the tip to substrate distance was 100 μm, which shows that the roughness was clearly on a much smaller length scale.

8. Since authors used different scale bar and color, it’s difficult to compare the images and results in Figure 5 with Figure 4, especially Figure 4A.

We agree with the reviewer: new and updated Figures 4 to 6 were prepared.

Reviewer #2:

Therefore, in my view, it is acceptable for publication after major revision taking into the account the following comments:

- The authors use the term “2D scanning electrochemical microscopy” throughout (e.g., line 19 in the abstract). It wasn’t clear to me what the 2D refers to? The title of the paper is also confusing. What does high throughput electrocatalysis mean?*

We agree with the reviewer and removed from the revised manuscript the term “2D SECM”. As an explanation, we originally used the term 2D scan to indicate that we did not carry out extensive studies as a function of the Z-coordinate (i.e., the 3-rd dimension), namely, the tip to substrate distance was fixed after the initial preliminary calibration.

Regarding the title, we also agree with the reviewer and we have simplified it. The new title is: *Application of Scanning Electrochemical Microscopy for Electrocatalytic Activity and Product Selective Screening of CO₂ Reduction Catalyst Arrays*

By high-throughput electrocatalytic screening we mean automated screening, at the same time and under the same conditions, of a large number (e.g., linear or rectangular arrays (matrices)) of electrocatalyst samples. This involves both special experimental techniques and machine learning based processing algorithms of large data sets. In scope, this endeavor is similar to high throughput screening of biological samples and, in our opinion, high throughput electrocatalyst screening, if it could be done reliably and properly validated, would represent a tremendous advancement for a wide range of important reactions.

- In the abstract (and conclusions), the authors allude to “product-selective detection”. One well-known limitation of amperometric detection (such as SG/TC SECM) is the lack of selectivity (also seen in the paper, as the SECM tip signal arises from both formate and H₂ oxidation). There are other product produced at the substrate such as CO. Is the tip potential of -0.5 V for SECM 2D scans selective to the formate? This needs to be rationalised further.*

We agree with the reviewer and all text and figures devoted to the detection at a constant potential of -0.5 V was removed, since it is not selective enough and only muddies the main points of this work. Therefore, in the revised MS the focus is on the fast CV detection method, which in our opinion, can be used for selective detection of the different products (formate, CO, H₂). This is emphasized in the revised Abstract, main body of the paper and Conclusions.

• Another known limitation of using a solid electrode probe to monitor an electrocatalytic process is that the SECM tip can be prone to “fouling”, meaning that the signal at the tip changes with time. This is shown in Figure S4, Figure S5, and is also seen Figure S6 (the response changes with time/cycling). This is likely to become more problematic with more complicated product detection (see “product-selective detection” comment above). How can such an effect be prevented/accommodated for?

The reviewer is right, constant potential detection does not work well and it is prone to tip fouling (poisoning). Therefore, it was entirely removed from the revised manuscript and the focus is only on the fast (1 V s⁻¹) CV detection which works well. To address the tip fouling we introduced a new Figure 5, to show that the starting potential for the fast CV makes a big difference in terms of scan stability over time. We chose 1.2 V vs Ag/AgCl as the preferred starting potential and we demonstrate the reproducibility of scans (new Fig. 5). Furthermore, during CV-SECM scan there is a rest period for 10 s at 1.2 V between two data acquisitions, and that period also serves a tip cleaning purpose. These aspects are all described in the revised manuscript.

• In Fig. 3C, why is there a difference between CO₂ and N₂ atmosphere? Furthermore, why is there a non-zero intercept?

N₂ (pH = 8.75) and CO₂ (pH = 6.75) saturated solutions provide a different solution pH, which plays a role in the HCOO⁻ oxidation mechanism on Pt electrodes. For this reason the oxidation current achieved is slightly different in both cases but a linear correlations between peak current density and concentration could be established for both conditions (Fig. 3C).

Additionally, within the normal experimental error for both conditions at zero formate concentration the current densities are virtually zero (there is a very small background current associated with double layer charging) (Fig. 3C).

• SECM is nontrivial to be used as a screening technique (i.e., low throughput per sample due to long experimental time) and it is more suitable to study kinetics and spatial heterogeneity of the catalyst in the fundamental aspect. The authors refer to the presented method as “high-throughput” throughout (e.g., Line 193). When reading through the Experimental section, it seems that a full set of experiments would take a few hours to carry out (e.g., the fast-scan CV scan takes 2.5 hours alone), so my question is: is this truly a high-throughput method? It seems like the equivalent set of experiments could be carried out a rotating ring-disk electrode (RRDE) on a similar timescale. The author should revisit the introduction and emphasise more specifically why the SECM is important for this work.

If we only show the first 5 scan lines of CV-SECM as in the new Figure 6, the imaging time is reduced to 75 min and we provide the same information/conclusions regarding the CO₂RR that we provided showing 10 scan lines. In addition to time and human resources savings compared to the conventional one-sample-at-a-time methods, there are other significant advantages as well for SECM based screening of samples under the same conditions simultaneously. For example, each manual replacement of the individual catalyst sample for a new RRDE test could introduce uncontrolled variabilities regarding impurities. Similar to biological sciences, there are tremendous advantages in trying to develop automated methods for screening in spite of the inherent difficulties. In our opinion in the present work we succeeded in laying some of the groundwork on which further advancements can be made toward extension of SECM in this direction.

• *On Line 200: “free from artifact due to tilt, roughness and resistivity of the substrate array”. Surface roughness can cause artifacts for most scanning probe methods and SECM is no exception. As real electrocatalysts tend to possess a degree of roughness (noted in Figs. S1 and S2), is this technique truly applicable with “non-model” substrates, as claimed in the introduction (lines 68 – 75)? Additionally, how can the authors be confident that an increase in current detected at the SECM is due to an increase in activity rather than just an increase in catalytic surface area?*

Substrate surface roughness is not relevant from a catalytic activity point of view if the reaction under study (CO₂RR) is taking place on the Sn/SnO_x electrode under diffusional control conditions, since only geometrical surface plays a role then. Reported SECM images in this manuscript are collected holding the substrate potential at –1.5 V vs Ag/AgCl, which means applying a relevant overpotential, which ensures diffusional control conditions. Furthermore, if it were only a surface area effect the best results for formic acid production in Fig. 7 would be obtained for the high surface nanoparticulate sample (i.e. prepared by pre-electroreduction at –3 V). This is clearly not the case, thus the intrinsic activity due to the surface oxide (SnO₂) content is very important.

• *In Fig. 4A, what is the y-contrast in the pre-reduced at -1.25 V sample (left-most plot)? It looks like the top of the sample possesses a different activity to the bottom. Is this an artifact caused by tip/substrate fouling?*

Figure 4 was replotted and a new Fig. 6 with fast CV mapping was added. Based on the reasons we explained in a previous answer regarding the start potential effect on the tip CV, we consider that none of our data suffers from tip/substrate fouling during the time frame of the experiments.

• *On line 219, the authors mention “catalyst ‘cross-talk’, plays no role...”, but in Fig. 4A, S7 and S8, the three catalysts do not appear to diffusionaly isolated, apparent from the non-zero current detected between the Sn/SnO_x strips.*

In the revised manuscript we produced a new Fig. 6 from the original Fig. S9 maintaining the first 5 scan lines. The new Fig 6 shows clearly the cross-diffusional isolation of the samples, in other words, the absence of catalyst ‘cross-talk’. Similarly, Figs. S5, S6 and S8 in the

Supplementary, demonstrate also the same diffusional isolation. Moreover, the current signal to noise ratio detecting products by fast CV is very high.

• Further to the points above, how feasible would it be to expand this method to a larger array of catalysts? My feeling is that artifacts from sample tilt, tip-fouling and diffusional cross-talk (see comments above) would ultimately limit the applicability of this technique.

For increasing the number of samples to be studied we will only need to reduce the size of the samples to maintain reasonably short imaging time. Moreover, smaller samples will be easier to be managed.

• On lines 220 – 223 “The 2D spatial variation within a sample can be attributed to a large degree to the non-homogeneous catalyst surface in terms of the distribution of SnOx and metallic Sn.” This is potentially a very important observation, as it indicates that the catalysts are not homogeneously-active. Is it possible to couple the SECM scanning with (ex situ) co-located compositional/structural analysis (e.g, XPS mapping) to understand the nature of the catalytic hotspots? Or is it possible that the apparently “high-activity” areas are simply more rough/porous, giving rise to a higher catalytic surface area?

Based on the information we have now it is very hard to draw reliable conclusions about surface inhomogeneities and hotspots affecting the electrocatalytic activity. However, as the reviewer suggested in future work it would be very beneficial to connect localized surface analysis (e.g. XPS) with CV-SECM scans. A sentence pointing to this has been added to the Conclusion.

• Reading through the Experimental section, the catalyst undergoes prolonged exposure to aerated 0.1 M KCl prior to scanning. Oxygen reduction is also performed on the catalyst surface (e.g., Fig. S7) prior to CO₂ electroreduction screening. How do these protocols affect the surface structure/composition of the electrocatalysts? Additionally, could surface contamination with chloride and/or oxy/hydroxy species be a contributor to the observed trends? (i.e., could the -3 V sample be more susceptible to contamination/fouling, leading to apparently lower activity?)

The sample is never immersed in solution at open circuit potential. So, the potential damage from Cl⁻ in solution, which might provoke corrosion on the Sn electrodes, is prevented.

Regarding the high surface area sample (produced at – 3V) we provide an explanation of its activity based on the low initial SnO₂ content coupled with enhanced reduction of the metastable oxide during CO₂ reduction forming metallic Sn, which is known to enhance the H₂ evolution reaction while suppressing CO₂RF (please see p.8).

• Main results in Figure 4 and Figure 5: The author needs to be more clear about the results from “SECM 2D scans with constant tip potential”. As the author addressed already, the Pt is contaminated with CO and SnO₂ and this significantly changes electrochemical activity of the probe (figure S4 and S5). As shown in the image in the Figure 4, the current density is two

orders of magnitude lower than what it should be when Figure 4A is compared to Figure 5B (as well as Peak 1 in Figure 3D). The author needs to validate what it is measures in the “SECM 2D scan with constant tip potential” and if it is directly relevant to formate detection.

We agree with the reviewer, and as mentioned earlier as well, all the figures and discussion relevant to the constant tip potential detection were eliminated from the revised manuscript. In the revised manuscript the focus is only on the fast (1 V s^{-1}) CV detection.

- *Minor comments: A lot of figure captions are lack of details of the figures in both main text and the supporting information. The author needs to ensure to provide all details of the figure to the readers.*

We improved the figure captions throughout the manuscript and supplementary material.

Reviewers' comments:

Reviewer #1 (Remarks to the Author):

Thank the reviewer for the effort. My concerns have been fully addressed, and I recommend the manuscript to be published at its present form.

Reviewer #3 (Remarks to the Author):

This is an interesting study focusing on the application of SECM for the detection of formate (and co-products, CO and H₂) during the CO₂ reduction on tin and tin-oxide-based catalysts. The authors have addressed most of the raised questions, but there are some important issues that should be solved:

Some questionable points should be revised by the authors. For example, authors stated: that "...the electrochemical reduction of SnO₂ to the less active metallic form at potentials relevant for CO₂RF_{5,6}, is a major drawback". However, it is well-known that SnO₂ can not be the active phase, since its electronic conductivity is too low. Why authors did not mention the formation of an "active" tin surface after electrochemical reduction of SnO₂? The formation of Sn atoms with very low coordination number (and number of grain boundaries) has been demonstrated to be a key point in several previous studies in literature;

The authors' explanation based on the removal and reformation of the SnO₂ layer upon exposure to ambient air is interesting, but the resulting activity may be associated to the different defects produced at the Sn surface (grain boundaries, undercoordinated Sn atoms..etc), which are known to activate the CO₂ molecules;

Faradaic efficiency is the main challenge and is the focus of CO₂ reduction studies. There is no quantitative analysis in the present study. In Page 8, authors mentioned that "...the wave associated with H₂ oxidation is completely lacking for the sample pre-electroreduced at -1.25 V indicating virtually 100% Faradaic efficiency for CO₂ electroreduction to carbonaceous products (formate and CO)". This conclusion has to be supported by quantitative analysis. (the H₂ oxidation wave overlaps that of the first peaks of formate oxidation during the CV positive-going scan).;

The size of Figure 6 should be increased.

Finally, this author does not see any reason for the publication of this study as a communication, since this topic of study (including electrochemical detections and SnO_x) has been published before (several works and several different authors).

Rebuttal Letter:

The authors greatly appreciated the insightful review of the manuscript. All the suggestions and recommendations were taken into account and, as detailed in the point-by-point answers below, appropriate changes and revisions were made, which are also highlighted in yellow in the revised manuscript.

Reviewer #3:

1. Some questionable points should be revised by the authors. For example, authors stated: that "... the electrochemical reduction of SnO₂ to the less active metallic form at potentials relevant for CO₂RR^{5,6}, is a major drawback". However, it is well-known that SnO₂ can not be the active phase, since its electronic conductivity is too low. Why authors did not mention the formation of an "active" tin surface after electrochemical reduction of SnO₂? The formation of Sn atoms with very low coordination number (and number of grain boundaries) has been demonstrated to be a key point in several previous studies in literature;

The nature of the active sites for CO₂ electroreduction has been intensely debated in the literature. The catalyst is a combination of metallic tin and tin oxide, therefore, similarly to many other oxide-based electrocatalysts used in electrochemistry (e.g., MnO₂, RuO₂, IrO₂), the electronic conductivity limitation of the thin oxide layer is not detrimental in practice, because it is combined with either the metallic form or other electronically conductive components (e.g., C support).

For CO₂ reduction to formate on Sn/SnO₂, it has been recognized in a number of publications that the oxide itself also plays a catalytic role by stabilizing the radical anion CO₂^{•-}. For the sake of brevity, here we refer only to the paper by S. Zhang, P. Kang, and T. J. Meyer. *J. Am. Chem. Soc.* 2014, 136, 1734-1737 (ref. 9 in our paper), indicating that the strong electron backdonation from the graphene support to the tin oxide is an important factor in the high activity. It has also been recognized that maintaining the stability of the surface oxide structure is important for the overall long term activity and durability (*J. Phys. Chem. C* **119**, 4884–4890 (2015), ref. 5 in our paper). We agree with the reviewer that the ‘nascent’ Sn formed *in situ* by oxide reduction can also serve as catalytically active sites. Therefore, we rewrote the relevant paragraph on p.2 to better reflect the possibility of different active sites (please see yellow highlights on p.2).

2. The authors explanation based on the removal and reformation of the SnO₂ layer upon explosion to ambient air is interesting, but the resulting activity may be associated to the different defects produced at the Sn surface (grain boundaries, undercoordinated Sn atoms..etc), which are known to activate the CO₂ molecules;

We agree with the reviewer that through this process of electrochemical stripping – atmospheric oxidation surface defects are likely to be introduced as well. Therefore, we rewrote the sentence on p.4-5 as follows:

'Furthermore, through this process of electrochemical stripping of the oxide followed by atmospheric oxidation it is likely that surface defects are also introduced that can also impact the electrocatalytic activity.'

3. Faradaic efficiency is the main challenge and are the focus on CO₂ reduction studies. There is no quantitative analysis in the present study. In Pag. 8, authors mentioned that "...the wave associated with H₂ oxidation is completely lacking for the sample pre-electroreduced at -1.25 V indicating virtually 100% Faradaic efficiency for CO₂ electroreduction to carbonaceous products (formate and CO)". This conclusion has to be supported by quantitative analysis. (the H₂ oxidation wave overlaps that of the first peaks of formate oxidation during de CV positive-going scan).;

It is not possible to carry out in situ GC analysis of gases from a specific sample from the catalyst array to determine the H₂ evolved. The electrochemical screening should be further validated by quantitative analysis but the only way to do that is to test and analyze each sample separately and collect the gases. However, herein we demonstrated that with fast CV scanning good resolution of the peaks can be obtained and an initial assessment of the activity map can be generated. In future work we plan to further validate this method by carrying out, among other tests, individual analysis by GC of the evolving gases (CO and H₂) and ion chromatography analysis for formate. These studies were beyond the timeframe and objective here and we will report on them in a future publication.

To address this point, in the revised manuscript on p. 9 we added:

In future work, separate ex situ validation of these electrochemical results will be sought, by performing flow cell experiments with down selected individual catalysts (e.g., prepared by pre-electroreduction at -1.25 V) coupled with complete quantitative analysis of gaseous and liquid products.

4. The size of Figure 6 should be increased.

We increased the size of Fig. 6 to fit the entire page.

5. Finally, this author does not see any reason for the publication of this study as a communication, since this topic of study (including electrochemical detections and SnOx) has been published before (several works and several different authors).

To our knowledge there is no other study that introduces the SECM technique for screening of an array of CO₂ catalysts. All previous studies focused on one catalyst analysis at a time. In this study we lay some of the groundwork for the application of SECM for screening catalyst arrays and matrices, which could lead to high-throughput screening and acceleration of the catalyst discovery process. We introduce the fast CV detection at the Pt ultramicroelectrode to

distinguish selectively the three products (formate, CO, H₂). Furthermore, another novelty is the finding that the Sn/SnO_x catalyst produced by the pre-electroreduction method at -1.25 V vs Ag/AgCl displays excellent intrinsic activity and it could lead to a practical catalyst that must be investigated further and in larger cells, e.g., flow cells. We also address the formate generation activity and we link those results to the presence of specific oxides on the Sn surface quantified by XPS.

Reviewer #3 (Remarks to the Author)

'Editorial note: this reviewer provided no further comments for the authors.'